# N-Glycomics of Human Erythrocytes

**DOI:** 10.3390/ijms22158063

**Published:** 2021-07-28

**Authors:** Rosaria Ornella Bua, Angela Messina, Luisa Sturiale, Rita Barone, Domenico Garozzo, Angelo Palmigiano

**Affiliations:** 1CNR, Institute for Polymers, Composites and Biomaterials (IPCB), 95126 Catania, Italy; angela.messina@cnr.it (A.M.); luisella.sturiale@cnr.it (L.S.); domenico.garozzo@cnr.it (D.G.); 2Laboratorio di Sanità Pubblica, Sezione Tossicologia, Azienda Sanitaria Provinciale (ASP), 95124 Catania, Italy; 3Child Neurology and Psychiatry, Department of Clinical and Experimental Medicine, University of Catania, 95123 Catania, Italy; rbarone@unict.it

**Keywords:** red blood cells, N-glycosylation, MALDI-TOF, ABO(H) blood groups

## Abstract

Glycosylation is a complex post-translational modification that conveys functional diversity to glycoconjugates. Cell surface glycosylation mediates several biological activities such as induction of the intracellular signaling pathway and pathogen recognition. Red blood cell (RBC) membrane N-glycans determine blood type and influence cell lifespan. Although several proteomic studies have been carried out, the glycosylation of RBC membrane proteins has not been systematically investigated. This work aims at exploring the human RBC N-glycome by high-sensitivity MALDI-MS techniques to outline a fingerprint of RBC N-glycans. To this purpose, the MALDI-TOF spectra of healthy subjects harboring different blood groups were acquired. Results showed the predominant occurrence of neutral and sialylated complex N-glycans with bisected N-acetylglucosamine and core- and/or antennary fucosylation. In the higher mass region, these species presented with multiple N-acetyllactosamine repeating units. Amongst the detected glycoforms, the presence of glycans bearing ABO(H) antigens allowed us to define a distinctive spectrum for each blood group. For the first time, advanced glycomic techniques have been applied to a comprehensive exploration of human RBC N-glycosylation, providing a new tool for the early detection of distinct glycome changes associated with disease conditions as well as for understanding the molecular recognition of pathogens.

## 1. Introduction

The red blood cell (RBC), or erythrocyte, is the simplest human cell, as it does not have internal organelles, which are lost during the erythropoiesis process. RBCs develop in the bone marrow and have a life span of about 100–120 days before they are recycled by macrophages [1]. In the last decade, several studies were carried out to shed light on the biological function of RBC membrane proteins [2,3]. As a result, the erythrocyte membrane is one of the best-known membranes in terms of structure, function, and associated genetic disorders [4,5,6]. The RBC membrane mediates transport functions and provides the erythrocytes with their structural features of resilience and deformability [7].

Glycosylation is a post-translational modification characterized by the covalent linkage of oligosaccharides moieties (glycans) to form glycoconjugates. Glycans are conjugated to asparagine (N-glycan) or serine/threonine (O-glycans) residues to form glycoproteins. Protein glycans play important roles in biological function/activity, protein folding, and molecular recognition [8]. Glycans are also recognized by glycan-binding proteins with lectin activity on opposing cells, and these glycan-ligand interactions are responsible for many biological activities, including the induction of the intracellular signaling pathway and recognition of pathogens such as influenza virus [8,9].

Changes in N-linked glycosylation have long been associated with disease development, and acquired glycan modifications have been described in multifactorial diseases such as cancer [10,11], inflammation [12,13], neurodegeneration [14,15], and genetic diseases [16,17].

Human RBC protein glycosylation has not been systematically examined in its entirety yet [18], though previous studies investigated the two most abundant glycoproteins of the erythrocyte membrane, namely band 3 and glycophorin A (GPA) [19,20,21,22]. In human erythrocytes, band 3 and other red blood cell membrane proteins bear poly-N-acetyllactosamine (poly-LacNAc) extensions, acting as an antigen carrier during various stages of cell development and differentiation [18,19]. The ABO(H) antigens were found as terminal non-reducing epitopes of large complex RBC N-glycans [23,24] and also constitute distinctive traits of outer arm glycophorin N-linked structures at lower molecular masses [18,25]. They play crucial roles in transfusion medicine, determining the compatibility between blood donors and recipients, as well as in the development of several hemostasis-related genetic disorders, including cardiovascular diseases and thrombosis [26,27]. Furthermore, several studies have investigated the relationship between the ABO(H) blood group distribution and the occurrence of some infectious diseases, such as HIV, malaria, influenza [28,29,30,31,32,33], and, recently, the COVID-19 pandemic [34,35,36,37].

Although information on the biological roles of glycan-binding proteins is still rising, challenges remain in detecting and profiling known and novel cellular glycans structures, due to the vast heterogeneity of possible glycoprotein isoforms depending on the glycan occupancy of protein glycosylation sites (macroheterogeneity) and on the variety of the attached glycans (microheterogeneity). Macroheterogeneity is related to the site availability, enzyme kinetics, and substrate concentrations that regulate the N-glycan precursor assembly and its subsequent transfer to the protein in cytosol and endoplasmic reticulum, whereas microheterogeneity is associated with the N-glycan processing occurring in endoplasmic reticulum and Golgi [17]. In this context, mass spectrometry (MS) techniques have demonstrated their capability for the characterization of unknown glycans and the high-throughput analysis of known glycan structures, allowing individual glycoforms to be distinguished [38]. Advances in mass spectrometry, and in particular in matrix-assisted laser desorption/ionization time-of-flight (MALDI-TOF) MS, have driven substantial progress in glycan analysis [39]. The eligibility of MALDI-TOF is established because of its ability to analyze complex mixtures of glycans from biological samples together with its high detection sensitivity, as exemplified by several glycomic studies [40,41,42].

Since RBCs are continuously replenished with glycosylated membrane proteins that can be isolated, purified, and readily available from blood samples, glycosylation analysis of erythrocyte membrane glycoproteins might be the key for the early detection of glyco-biomarkers for diagnostic and therapeutic purposes as well as for studying interaction mechanisms involved in recognition of pathogens. In the present study, we applied high-sensitivity MALDI TOF MS-based glycomic methodologies to the analysis of total N-glycans derived from human erythrocyte membrane proteins, with the aim to generate an N-glycosylation fingerprinting of RBC. In particular, the total N-glycome of healthy subjects with different ABO(H) blood groups is evaluated as the starting point for future investigations.

## 2. Results

### 2.1. RCB N-Glycoprofiles by MALDI-MS

The representative N-glycan profiles of membrane glycoproteins from human erythrocyte (blood groups A, B, AB and O) are reported in Figure 1A–D and in Appendix A. Each MALDI mass spectrum is very dense with peaks, showing some hundreds of assigned glycoforms which correspond to a wide range of N-Glycan categories, including a full complement of oligomannose structures and a series of complex glycans, mainly with bisecting N-acetylglucosamine (GlcNAc), as well as neutral, acidic, and hybrid species. All the acquired mass spectra were found to be very similar, with the distinguishing factor being the blood group epitope-bearing structures. In particular, the low mass-range of each spectrum (Figure 1A–D) is very informative as it comprises N-glycan structures with none, one, or both of the ABO(H) antigens. Below is a detailed description of the main N-glycan families recognized in the RBC N-glycome fingerprinting. Different signals in blood group A and B spectra (Figure 1A,B) differ by 41 u.m.a., the mass difference existing between N-acetylglucosamine and galactose. These peculiar structures of blood groups A and B are highlighted with red and blue circles, respectively (e.g., 2908.5, 3269.5, 3357.7, etc., for blood group A and 2867.4, 3228.6, 3316.7, etc., for blood group B).

#### 2.1.1. Oligomannose N-Glycans

Oligomannose N-glycans structures were observed at m/z 1579.8 (Man5), 1783.9 (Man6), and 2396.2 (Man9), as reported in Figure 1A–D and in Appendix A.

#### 2.1.2. Hybrid N-Glycans

Minor amounts of hybrid glycans were also detected at m/z 2173.1, corresponding to structures with core and peripheral fucosylation, and at m/z 2186.1 and 2390.2, attributable to sialylated species (Figure 1A–D and Appendix A).

#### 2.1.3. Bisected N-Glycans

Amongst the N-glycan structures shown in Figure 1A–D and in the Appendix A, bisected N-glycans were the most abundant species. The main ion peaks at m/z 2850.4 and 3211.6 corresponded to fucosylated mono and disialo-biantennary oligosaccharides with a bisecting GlcNAc residue, as reported by previous studies [21,22,25]. Bisecting GlcNAc is a common structural feature of erythrocyte N-glycans, together with antennary fucosylation and a significant sialylation level [19,20,21,22,25,43,44]. Most of the bisected N-glycans identified here showed core fucosylation and poly-LacNAc branching.

#### 2.1.4. Poly-LacNAc N-Glycans

In the middle/high mass regions of the spectra (shown in the respective Appendix A), N-glycans bearing poly-LacNAc extensions (Galβ1-4GlcNAcβ1-3-) were observed, with the largest glycans possessing about a dozen LacNAc repeats. It is reasonably expected that large structures with more than two LacNAc extensions may carry the major portion of ABO(H) antigens as terminal epitopes, producing a very complex typical fingerprint of each blood group.

#### 2.1.5. Sialylated N-Glycans

In our MALDI MS spectra, sialylation was a relatively abundant terminal decoration, especially in the higher mass regions, with the largest glycans bearing between one and three sialic acid (NeuAc) terminated epitopes on their antennae.

#### 2.1.6. ABO(H) Antigen-Bearing N-Glycans

Figure 1A reports a typical RBC N-glycan MALDI spectrum for blood group A focused on the mass-range between m/z 1500 and 4000. The related epitope [GalNAcα1-3(Fucα1-2)Galβ1-4GlcNAc-] was found as a terminal side chain substitution of the glycoforms at m/z 2908.5, 3269.6, 3357.7, 3531.8, 3718.9, 3806.9, and 3981.0 (Figure 1A and Appendix A), as demonstrated by further in-depth structural characterizations by MALDI TOF/TOF MS/MS. Moreover, at over m/z 4000, a number of molecular ions was found consistent with bisected N-glycans bearing poly-LacNac elongations mostly terminating with blood group A antigen or NeuAc, as shown in Appendix A. Many of the assigned structures were confirmed by MS/MS analysis. As a representative example, the fragmentation spectrum of the molecular ion at m/z 2908.5 is reported in Figure 2.

It comprises intense B and Y ions originating from glycosidic linkage cleavages (Domon and Costello nomenclature [45]), mainly present on permethylated glycan fragmentation patterns. In particular, the non-reducing terminal A antigen of the molecular ion at m/z 2908.5 gave rise, in the low mass-range, to a series of B ions at m/z 660.2, 905.4, and 1109.7, together with internal fragments at m/z 646.3 (BY ion) and 735.4 (CY ion), eliciting in the high mass-range a very intense Y fragment at m/z 2026.0 due to the blood group A epitope loss from the parent ion.

Likewise, the B antigen [Galα1-3(Fucα1-2)Galβ1-4GlcNAc-] was found as a side chain substitution of the glycoforms at m/z 2867.4, 3228.6, 3316.7, 3490.8, 3677.8, 3765.9, and 3940.0 (Figure 1B and Appendix A) and was coherently suggested as the terminal structural motif of a number of high-mass glycans up to m/z 8082.0 (see highlighted structures in Appendix A). Figure 3 shows the MS/MS spectrum of a typical B antigen-bearing N-glycan structure, i.e., the ion at m/z 3677.8 present in Figure 1B and Appendix A.

The fragmentation pattern comprised a series of peaks demonstrating the occurrence of a B antigen as B-type ions at m/z 619.2, 864.3, and 1313.7 and the internal fragment at m/z 3200.5 (YY ion), revealing the loss of terminal unsubstituted galactose (Gal) and GlcNAc units, and the intense Y ion at m/z 2836.5 originated from the loss of blood group B epitope from the parent ion.

The RBC N-glycan MS analysis of blood type AB is reported in Figure 1C and in Appendix A. These spectra comprise both N-glycans carrying A or B antigens, originating couples of peaks with m/z 41, due to the mass shift between the correspondent permethylated structures bearing group A or group B epitopes (i.e., m/z 2867.4 and 2908.5, 3228.6 and 3269.6, 3316.7 and 3357.7, and 3677.8 and 3718.9 in Figure 1C), also found up to m/z 7761.9 (see Appendix A). As expected, MS/MS analyses of those N-glycans carrying specific antigens in AB group MS profiles yielded identical results as the respective A and B antigen-bearing ions detected in A or B blood group RBC samples (data not shown).

Figure 1D finally shows the N-glycan MS profile at a low mass-range (m/z 1500–4000), representative of the O (H) blood group RBC. N-glycans with group A or B epitopes are absent here, so MS profiles from group O can be considered as a control sample. An increased intensity of molecular ions consistent with glycoforms carrying the blood group H epitope (Fucα1-2Galβ1-4GlcNAc) was observed at m/z 2663.3, 3024.5, 3286.7, 3561.8, and 3923.0 (see also Appendix A). Blood group O individuals have functionally inactive A/B glycosyltransferases, and therefore their H antigen, the starting substrate for distinctive A/B blood group epitopes, remains unmodified [46]. As a consequence, the whole MS profile (as also reported in Appendix A) is characterized by the lack of antigen-bearing N-glycans with unique molecular masses. MS/MS analyses of the species at m/z 2663.3 and 3923.0 are presented as significant examples. The ion at m/z 2663.3 (Figure 4) gave a fragmentation spectrum characteristic of both core and antennary fucosylation. In particular, the B/Y ion pairs at m/z 864.4/1821.9 and 660.3/2026.0, together with the internal fragment at 432.2 (H+ form), implying the alternative location of the peripheral fucose either on terminal Gal and on antennary GlcNAc residue, suggested the possible occurrence of either H antigen or Lewis epitope as the terminal side chain substitution.

Similar results were obtained when analyzing the species at m/z 3923.0 (Figure 5), corresponding to a possible mixture of isobaric species with mono- and tri-LacNAc extensions at the two branches which may be alternatively capped with NeuAc or antigen H/Lewis epitopes. Further fragmentation analyses of molecular ions at m/z 3024.5, 3286.7, and 3561.8 concurred to postulate the co-presence of N-glycan isomers bearing H or Lewis antigens (data not shown).

## 3. Discussion

In the present study, we applied a robust and widely used MS-based glycomic approach to the extensive N-glycan characterization of RBCs. Our MALDI-TOF data showed erythrocyte membrane glycoproteins holding in prevalence complex bisected N-glycans with core and antennary fucosylation and recurrent poly-LacNAc extensions, in accordance with earlier glycosylation studies on specific red blood cell membrane proteins. The previous characterization of RBCs’ N-glycoproteins has been limited to their main components, such as band 3 and GPA proteins [19,20,21,22,25]. Band 3 was found harboring a single N-linked oligosaccharide, with a branched structure varying in the number of repeating LacNAc units terminated with Gal, fucose (Fuc), or NeuAc [19,20], whereas GPA, a major glycophorin, has been reported to bear 15 O-glycans [47,48] and a single N-linked glycan, mostly a biantennary sialylated moiety with bisecting GlcNAc and outer arm fucosylation [21,22]. In the current study, the MALDI-TOF strategy allowed a broad characterization of the total RBC N-glycans released from band 3, GPA, and additional minor glycoproteins. We observed large N-glycan structures up to 9 kDa (see Appendix A) as a result of the ad hoc developed MS strategy that led to a notable improvement in upper mass-range sensitivity and signal-to-noise ratio. Besides a predominant portion of complex highly processed structures, we also found oligomannose and hybrid N-glycans. As glycan structures are generated in the compartmentalized Golgi, changes in the relative signals of all the observed RBC N-glycans could be used as a diagnostic tool for the detection of defects in glycosylation enzymes involved in early Golgi processing in glycosylation-related diseases [43,49,50,51]. However, only a few studies reported on the N-glycosylation of human erythrocyte membrane glycoproteins using MS techniques [43,44,52], mostly focusing on the characterization of glycans from band 3 membrane glycoprotein in congenital dyserythropoietic anemia type II (CDA II), also called hereditary erythroblastic multinuclearity with the positive acidified-serum test (HEMPAS) [43,44]. Fukuda et al. in 1987 developed a method based on fast-atom bombardment (FAB) MS [43], whereas Denecke et al. in 2008 [44] compared erythrocyte band 3 mass mapping from HEMPAS and from a control by MALDI-TOF MS following SDS-PAGE and lectin-binding strategies. Both these studies accordingly found the lack of the large oligosaccharide component bearing the poly-LacNAc branches and the prevalence of glycans at lower molecular mass (such as oligomannose and hybrid and truncated complex species) in the red blood cell band 3 glycoprotein from HEMPAS patients, suggesting a defective Golgi processing in erythroblasts [43,44].

RBC membrane N-glycans are particularly exposed to the external environment, supporting the pathogen recognition processes. For instance, the hemagglutination assay is based on the interaction between the hemagglutinin located on the surface of the human-adapted influenza virus and some specific sialylated glycans on the epithelial cells of the human upper respiratory tract, defined as the key initial step of the infection cycle [53]. Accordingly, agglutination of chicken RBCs (cRBCs) [54] has long been used in viral titer assays as well as to investigate glycan receptor binding sites and in the testing of vaccines’ effectiveness [55,56,57,58]. The structural characterization of cell surface N-glycans of cRBCs, revealed the presence of bi- and triantennary structures capped with both α2→3 and α2→6 linked NeuAc and the lack of lactosamine repeating units [54]. On the other hand, the human bronchial epithelial cells, which are the target of human-adapted influenza A viruses, show the predominance of α2→6 sialylated glycans with lactosamine repeats [59,60]. These data could explain some pitfalls of the agglutination assay based on cRBCs which may not be representative of the physiological receptor for human-adapted influenza strains. Our study may trigger future advanced MS-based structural analyses on glycans from human RBCs, providing important insights for improved applications in this field.

The applied MALDI-TOF MS and MS/MS strategy presented here allowed for the characterization of glycans bearing the ABO(H) blood group antigens which, like genetic factors, are involved in several hemostasis-related diseases [26,27] and in many infectious diseases [28,29,30,31,32,33]. Several shreds of evidence suggest that the ABO(H) blood group expression may influence the development and the progression of cardiovascular disease, thrombosis, and hemostasis disorders. In the last year, there has been a growing interest in studying the association between the ABO(H) blood group distribution and the dynamics of the COVID-19 pandemic [34,35,36,37]. Recently, Liu et al. [36] found a positive correlation between the occurrence of COVID-19 infection cases and the proportion of blood group A by analyzing the data from the official WHO database. These results agree with other previous studies [34,35], strongly suggesting a relationship between the distribution of blood groups and the SARS-CoV-2 infection. However, the underlying mechanisms have not yet been clarified and further investigation, also taking into account the glycan-binding specificity and the glycosylation features of the SARS-CoV-2 proteins, is highly needed.

## 4. Materials and Methods

Unless otherwise noted, all reagents were purchased from Merck Chemicals GmbH, Darmstadt, Germany and were of the highest purity available.

### 4.1. Erythrocyte Isolation

Peripheral whole blood samples in anticoagulant tubes were obtained from healthy subjects who underwent blood type analyses. Procedures were performed according to Helsinki declaration after obtaining written informed consents from all participants. The erythrocytes were isolated by centrifugation for 10 min at 4000 RPM (equal to 2600 RCF× *g*) in a DuPont Sorvall RC-5B refrigerated centrifuge equipped with a GSA rotor (Du Pont Company Medical Products Sorvall^®^ Instruments, Wilmington, DE, U.S.A.). The supernatant, composed of plasma and buffy coat (leukocyte and platelets), was removed, and the cell pellet was washed three times with 3 mL of 0.9% saline solution.

### 4.2. Hemolysis and Extraction of Membrane Proteins

Erythrocyte membranes were prepared by slight modifications of a previously reported protocol [61]. Erythrocytes (500 µL) were suspended in an equal volume of saline solution, and as such constituted the erythrocyte suspension. Hemolysis was performed by adding 10 volumes of ice-cold hypotonic phosphate buffer (1.43 mM NaH2PO4, 5.7 mM Na2HPO4, pH 7.4). This mixture was gently stirred for 45 min at 4 °C, and the membranes were settled by centrifugation for 12 min at 12,000 RPM (23,400 RCF× *g*). The supernatant was carefully removed, and the pellet was washed 5–6 times with 10 volumes of hypotonic phosphate buffer till a white ghost appeared. Then, the supernatant was removed, and the ghosts were lyophilized. RBC membrane glycoprotein extractions were performed by Rapigest^TM^ surfactant (Waters Corporation, Milford, MA, U.S.A.) as a denaturing agent. About 5 mg of erythrocyte membranes were resuspended in 220 µL of Rapigest^TM^ 0.1% in 50 mM bicarbonate buffer, and the obtained mixture was boiled at 100 °C for 7 min. After cooling, samples were reduced in 5 mM dithiothreitol (DTT) at 56 °C for 30 min and alkylated in 15 mM iodoacetamide (IAA) in the dark at room temperature for 30 min. The N-linked glycans were released by 4 U of peptide N-glycosidase F (PNGase F, EC 3.5.1.52; Roche Diagnostics GmbH, Mannheim, Germany) digestion at 37 °C overnight. The enzymatic digestions were stopped by adding HCl (to a final concentration of 40 mM, pH ≤ 2) and incubated at 37 °C for 45 min. After dilution with MilliQ water (Millipore Simplicity^®^ UV, Millipore, Bedford, MA, U.S.A.) to a final volume of 1 mL, samples were centrifuged 10 min at 13,000 RPM (27,500 RCF× *g*), and the undissolved material was removed. The obtained N-glycans were purified by 3 cc C18 Sep-Pak cartridges (Waters Corporation, Milford MA, U.S.A.), followed by a further purification step by solid-phase extraction (SPE) (Hypersep Hypercarb, Thermo Fisher Scientific Inc., Waltham, MA, U.S.A.), and finally permethylated as described previously [62,63] to enhance mass spectrometric detection sensitivity.

### 4.3. MALDI TOF MS and MS/MS Analysis

A few microliters of permethylated N-glycan samples, resuspended in methanol, was mixed with the same volume of matrix solution (10 mg/mL 5-Chloro-2-mercaptobenzothiazole, CMBT in MeOH/H2O 80/20). MALDI TOF and MALDI TOF/TOF mass spectra were recorded in reflectron mode and positive polarity using a 4800 MALDI TOF/TOF™ (Applied Biosystems, Foster City, CA, U.S.A.) instrument, equipped with an Nd:YAG laser at 355-nm and 200 Hz repetition rate. In MS mode, 1200 shots were accumulated for each spectrum, with a resolution greater than 15K and a mass accuracy better than 75 ppm. A 4700-calibration standard kit, calmix (Applied Biosystems, Foster City, CA, U.S.A.), was used as external calibrant for the MS mode, and [Glu^1^] fibrinopeptide B human was used as external calibrant for the MS/MS mode (1 μL of TOF/TOF Calibration Mixture in 24 μL of CHCA matrix solution).

### 4.4. Data Evaluation

Data were processed using DataExplorer^®^ v.4.9 (build 115, Applied Biosystems, Foster City, CA, U.S.A.). Structural assignments were based on molecular weight identification, on the knowledge of the N-glycan biosynthesis, and on MS/MS analysis of specific glycoforms, when possible. N-glycan species were identified by bioinformatics such as Expasy GlycoMod (http://web.expasy.org/glycomod/ accessed on 5 April 2021), Glycoworkbench v2.1 [64], and the Consortium for Functional Glycomics glycan structures central database. Knowledge of the taxonomy and the exact mass (within 50 ppm) are often sufficient to define one or two structures (very rarely more than two). In these cases, a MALDI TOF/TOF MS/MS spectrum is able to establish which isomer it is or if we are in the presence of a mixture. In the spectra reported in this manuscript, we have drawn a structure when through this methodology we have identified a single structure, while we have used braces to indicate multiple structures. All this is in accordance with the SNFG approved cartoon representation.

## 5. Conclusions

Our developed MS strategy led to a considerable improvement in upper mass-range sensitivity and in signal-to-noise ratio, in addition to a significant increase in the resolution of MALDI-TOF mass spectra, allowing for a detailed mapping of human RBC N-glycans. Since RBCs have a relatively short lifespan, these analytical strategies could be used to study possible glycosylation changes that can occur during disease conditions, for the early detection of potential glyco-biomarkers. Most important, this developed strategy could be a useful tool to investigate the interaction mechanisms of pathogen recognition as well as the ABO(H) blood group-mediated response to viral infections, with special regard to SARS-CoV-2.

## Figures and Tables

**Figure 1 ijms-22-08063-f001:**
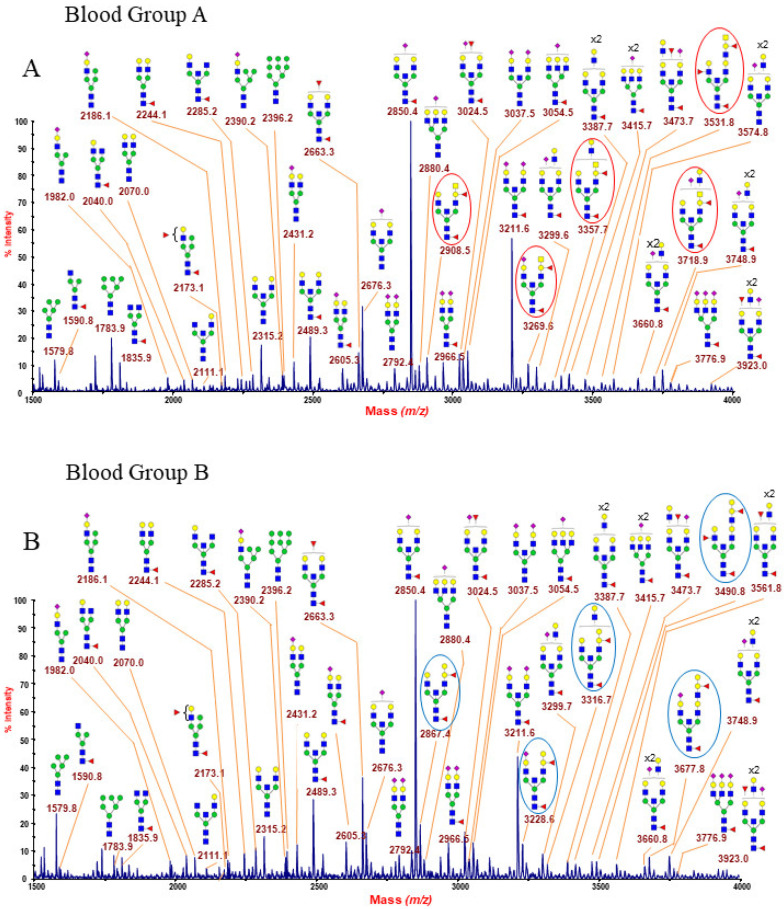
(**A**,**B**). MALDI-TOF mass spectra at low mass-range (m/z 1500–4000) showing molecular ions [M + Na]^+^ of permethylated N-linked glycans from human erythrocyte blood groups A and B. MALDI-TOF MS mapping of permethylated N-glycans from RBC is distinctive of each blood group. (**A**) Blood group A; (**B**) blood group B. Structures consistent with glycoforms comprising blood group A and blood group B epitopes are highlighted with red and blue circles, respectively. N-acetylglucosamine (GlcNAc), blue square; Mannose (Man), green circle; Galactose (Gal), yellow circle; Sialic acid (NeuAc), purple lozenge; Fucose (Fuc), red triangle. (**C**,**D**). MALDI-TOF mass spectra at low mass-range (m/z 1500–4000) showing molecular ions [M + Na]^+^ of permethylated N-linked glycans from human erythrocyte blood groups AB and O. MALDI-TOF MS mapping of permethylated N-glycans from RBC is distinctive of each blood group. (**C**) Blood group AB; (**D**) blood group O. Structures consistent with glycoforms comprising blood group A, blood group B, and blood group O epitopes are highlighted with red, blue, and green circles, respectively. N-acetylglucosamine (GlcNAc), blue square; Mannose (Man), green circle; Galactose (Gal), yellow circle; Sialic acid (NeuAc), purple lozenge; Fucose (Fuc), red triangle.

**Figure 2 ijms-22-08063-f002:**
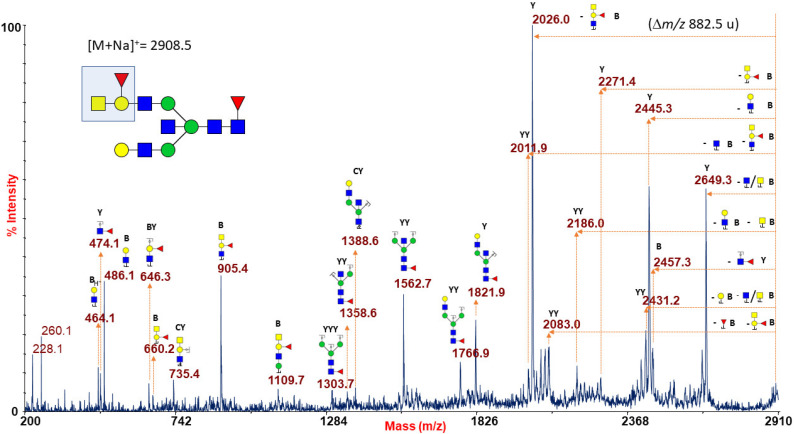
MALDI TOF/TOF MS/MS characterization of the N-glycan precursor at m/z 2908.5 from blood group A human erythrocytes. All the fragmentation pattern points to delineate an N-linked glycan moiety with a terminal blood group A epitope (sketched in the reported structure), as mainly revealed by B-type ions at m/z 660.2, 905.4 and 1109.7, by the predominant Y ion at m/z 2026.0 (m/z 882.5 u from the parent ion), and by a further Y ion at m/z 2271.4. GlcNAc, blue square; Man, green circle; Gal, yellow circle; NeuAc, purple lozenge; Fuc, red triangle.

**Figure 3 ijms-22-08063-f003:**
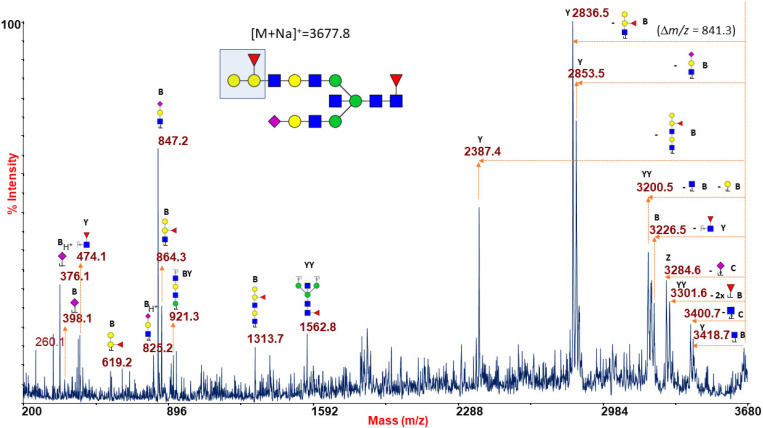
MALDI TOF/TOF MS/MS characterization of the N-glycan precursor at m/z 3677.8 from blood group B human erythrocytes. All the fragmentation pattern points to delineate a N-linked glycan moiety with a terminal blood group B epitope (sketched in the reported structure), as mainly revealed by B-type ions at m/z 619.2, 864.3, and 1313.7; by the predominant Y ion at 2836.5 (m/z 841.3 u from the parent ion); and by a further Y ion at m/z 2387.4. GlcNAc, blue square; Man, green circle; Gal, yellow circle; NeuAc, purple lozenge; Fuc, red triangle.

**Figure 4 ijms-22-08063-f004:**
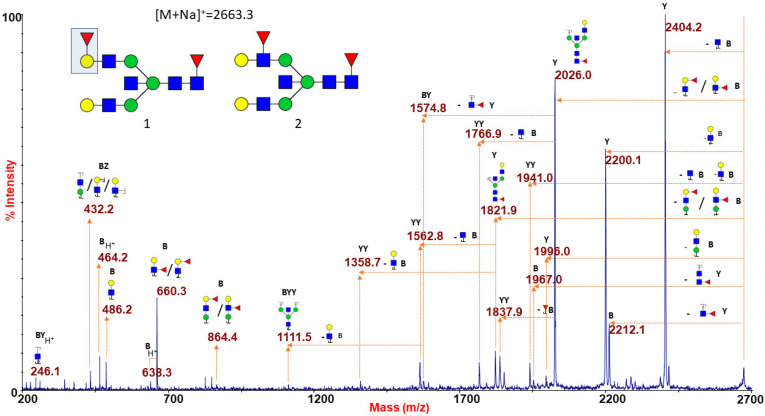
MALDI TOF/TOF MS/MS characterization of the N-glycan precursor at m/z 2663.3 from blood group O human erythrocytes. The reported spectrum shows the occurrence of two distinct isomers differing in the position of the antennary fucose. Fragments at m/z 660.3 and 864.4 (B ions) and at m/z 2026.0 and 1821.9 (Y ions) indicated that this monosaccharide may be linked either to a terminal Gal, giving rise to a blood group H antigen (as sketched in structure 1), or to an antennary GlcNAc, thus originating a Lewis epitope (structure 2). GlcNAc, blue square; Man, green circle; Gal, yellow circle; NeuAc, purple lozenge; Fuc, red triangle.

**Figure 5 ijms-22-08063-f005:**
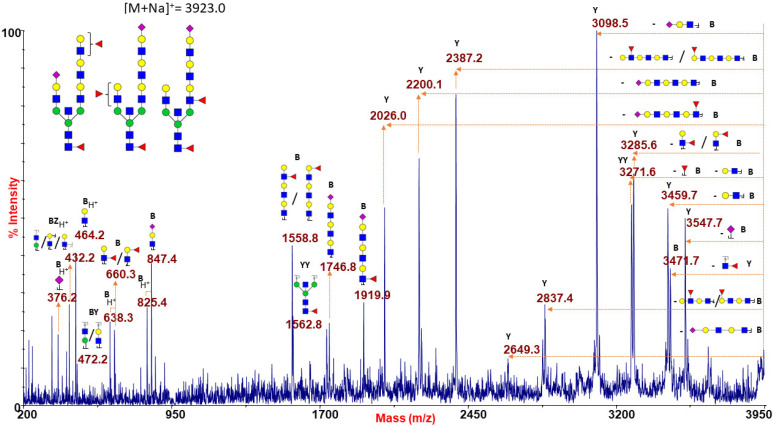
MALDI TOF/TOF MS/MS analysis of the N-glycan precursor at m/z 3923.0 from blood group O human erythrocytes. This fragmentation spectrum shows B and Y ion series revealing the co-presence of different isoforms characterized by mono- to tri-LAcNAc extensions which may be terminated with NeuAc or H antigen/Lewis epitopes. GlcNAc, blue square; Man, green circle; Gal, yellow circle; NeuAc, purple lozenge; Fuc, red triangle.

## Data Availability

All the MALDI TOF mass spectra and the MS/MS spectra recorded of permethylated N-glycans released by PNGase F can be downloaded from: www.ipcb.ct.cnr.it/ct/spectraVisualizza.jsp (accessed on 27 July 2021).

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
