# Peer review of "N-Glycomics of Human Erythrocytes"

_ijms, 2021, doi:10.3390/ijms22158063_

Round 1

Reviewer 1 Report

General comment:

This manuscript, entitled “N-Glycomics of human erythrocytes,” authored by Bua et al., reports exploring the human RBC N-glycome by high-sensitivity MALDI-MS techniques to outline a fingerprint of RBC N-glycans. This kind of approach is widely used in proteomics for connecting the biochemical abundance of signature molecules and is nice to answer fundamental biological questions in protein interactome. Diversity in glycoproteins makes these studies very complicated; however, high-resolution sensitive MS may be helpful tools in glycomics. Authors are also claiming  - for the first time, advanced glycomic techniques have been applied to a comprehensive exploration of human RBC N-glycosylation, providing a new tool for the early detection of distinct glycome changes associated with disease conditions as well as to understand pathogens molecular recognition. In my opinion, this is a valuable work and is suitable for publication in Int J Mol Sci after the authors have addressed the following comments and questions:

Specific questions:

  • Did the authors check any samples from a patient with a disease having potential differences in glycomes?
  • What are the differences in N-linked glycans from human erythrocytes blood groups A and B? I can see the similarity; can you mark the differences?
  • How can the author say that there are no false-positive results- did you run any control?
  • What are the complementary techniques one can use to prove these MS results?
  • Whether this technique can extract different modifications on different proteins of RBCs without doing any protein purification? If yes, please separate those modifications based on proteins (protein band 1- 7 and glycophorin alpha - delta)

Author Response

This manuscript, entitled “N-Glycomics of human erythrocytes,” authored by Bua et al., reports exploring the human RBC N-glycome by high-sensitivity MALDI-MS techniques to outline a fingerprint of RBC N-glycans. This kind of approach is widely used in proteomics for connecting the biochemical abundance of signature molecules and is nice to answer fundamental biological questions in protein interactome. Diversity in glycoproteins makes these studies very complicated; however, high-resolution sensitive MS may be helpful tools in glycomics. Authors are also claiming  - for the first time, advanced glycomic techniques have been applied to a comprehensive exploration of human RBC N-glycosylation, providing a new tool for the early detection of distinct glycome changes associated with disease conditions as well as to understand pathogens molecular recognition. In my opinion, this is a valuable work and is suitable for publication in Int J Mol Sci after the authors have addressed the following comments and questions:

 We are pleased to have the overall approval of our work and appreciate the points made by this reviewer for improvement of the manuscript.

Specific questions:

  • Did the authors check any samples from a patient with a disease having potential differences in glycomes?

We thank the reviewer for this very good question. Indeed, more than a question this is a suggestion for future developments of our work. This study is based on subjects who were not diagnosed with any disease. We usually study glycome differences in pathological versus normal control samples. The present study tested a novel, high-throughput method for RBC N-glycome analysis that results amenable for application to patients such as CDG, or other diseases, patients in future works.

  • What are the differences in N-linked glycans from human erythrocytes blood groups A and B? I can see the similarity; can you mark the differences?

The differences between group A and group B are due to the presence of two different epitopes: some signals in the spectra relating to epitopes A and B differ by 41 Dalton, the mass difference existing between N-acetylglucosamine and galactose. In Fig 1 A-B the peculiar structures of blood group A and B are highlighted by red and blue circles, respectively. This is now underlined in the text at page 3 lines 104-108.

  • How can the author say that there are no false-positive results- did you run any control?

This is really a good question, the purpose of this work anyway is not to make a direct comparison between patients affected by the disease and control samples, so it is hard to define  False Discovery Rate or false-positive results. However, it is possible to indicate the group 0 samples as the control samples as they do not have the characteristic epitopes of group A and / or group B.  Specific signals of group A or group B are not observed in the spectra relating to group 0. The text was amended accordingly at page 7 lines 211-212

  • What are the complementary techniques one can use to prove these MS results?

These results can be confirmed by complementary techniques such as NMR, LC-FLD with exoglycosidases digestions, Capillary Electrophoresis, etc.

  • Whether this technique can extract different modifications on different proteins of RBCs without doing any protein purification? If yes, please separate those modifications based on proteins (protein band 1- 7 and glycophorin alpha - delta)

We appreciate very much the point raised. The present study aims to analyze the total content of RBC N-Glycome from RBC membrane glycoproteins. We are already thinking about the development of methods to characterize the glycosylation of specific erythrocyte membrane proteins.

Submission Date

06 July 2021

Date of this review

19 Jul 2021 18:33:36

Reviewer 2 Report

In the present work, Bua et al. report on N-glycosylation profiles of ABO-serotyped human erythrocyte samples. In the course of their study, the authors meticulously analysed PNGaseF-released, permethylated N-glycans derived from human red blood cells using positive ion-mode MALDI-MS, and provide highly interesting insight into the structural features of select N-glycan compositions (critical to the ABO blood group system) by careful annotation of the respective MS/MS data. The paper is scientifically sound, very well written and of great interest to the fields of glyco-immunology. I recommend publication with very minor revisions.

Minor Comments:

In section 4.4 (line 340-346) the authors give some details on their Data Evaluation. Although the authors approach is well in line with current state-of-the art, it would be interesting to the reader to also have some numbers on e.g. the number of potential/known structural isomers “encoded” by the individual (most important) N-glycan compositions investigated in this study.

Line 45 “Glycans are also recognized by glycan-proteins with lectin activity [...]” – This reads awkward. “Glycoproteins with lectin properties” or “lectins”?

As “rpm” merely indicates the speed at which a given centrifuge rotor is operated, it is good practice to either also indicate centrifuge model/type or to provide the actual g-forces applied (i.e. value x g):

Line 301: “The 301erythrocytes were isolated by centrifugation for 10 min at 4000 rpm.”

Line 310: “This mixture was gently stirred for 45 min at 4 °C and the membranes were settled by centrifugation for 12 min at 12000 rpm.”

Line 323: “After dilution with MilliQ water to a final volume of 1 mL, samples were centrifuged 10 min at 13000 rpm and the unsolved material was re-324moved.”

Author Response

Minor Comments:

In section 4.4 (line 340-346) the authors give some details on their Data Evaluation. Although the authors approach is well in line with current state-of-the art, it would be interesting to the reader to also have some numbers on e.g. the number of potential/known structural isomers “encoded” by the individual (most important) N-glycan compositions investigated in this study.

Knowledge of the taxonomy, and the exact mass (within 50 ppm) are often sufficient to define one or two structures (very rarely more than two). In these cases, a MALDI TOF/TOF MS/MS spectrum is able to establish which isomer it is or if we are in the presence of a mixture. In the spectra reported in this manuscript, we have drawn a structure when through this methodology we have identified a single structure, while we have used braces to indicate multiple structures. All this in accordance with the SNFG approved cartoon representation. The text has been amended according at page 11 lines 361-367.

Line 45 “Glycans are also recognized by glycan-proteins with lectin activity [...]” – This reads awkward. “Glycoproteins with lectin properties” or “lectins”?

We are sorry for the imprecision; the sentence can be rephrased with “glycan-binding proteins with lectin activity” [Ref 8]. This has been corrected (page 1 lines 44-45)

As “rpm” merely indicates the speed at which a given centrifuge rotor is operated, it is good practice to either also indicate centrifuge model/type or to provide the actual g-forces applied (i.e. value x g):

Line 301: “The 301erythrocytes were isolated by centrifugation for 10 min at 4000 rpm.”

Samples were centrifuged in a DuPont Sorvall RC-5B refrigerated centrifuge at 4000 RPM equal to 2600 RCF x g (2602.2) The text was amended accordingly at page 10, lines 313-314

Line 310: “This mixture was gently stirred for 45 min at 4 °C and the membranes were settled by centrifugation for 12 min at 12000 rpm.” 23400 x g (23419.5). The text was amended at page 11, line 326.

Line 323: “After dilution with MilliQ water to a final volume of 1 mL, samples were centrifuged 10 min at 13000 rpm and the unsolved material was re-324moved.” 27500 x g (27485.3). The text was amended at page 11 line 339.

Submission Date

06 July 2021

Date of this review

12 Jul 2021 15:16:00 
